# Lead halide perovskite vortex microlasers

Wenzhao Sun[1,4], Yilin Liu[1,4], Geyang Qu [1], Yubin Fan [1], Wei Dai[1], Yuhan Wang[1], Qinghai Song [1,2], Jiecai Han[3] & Shumin Xiao [1,2,3✉]

Lead halide perovskite microlasers have been very promising for versatile optoelectronic applications. However, most perovskite microlasers are linearly polarized with uniform wavefront. The structured laser beams carrying orbital angular momentum have rarely been studied and the applications of perovskites in next-generation optical communications are thus hindered. Herein, we experimentally demonstrate the perovskite vortex microlasers with highly directional outputs and well−controlled topological charges. High quality gratings have been experimentally fabricated in perovskite film and the subsequent vertical cavity surface emitting lasers (VCSELs) with divergent angles of 3º are achieved. With the control of Archimedean spiral gratings, the wavefront of the perovskite VCSELs has been switched to be helical with topological charges of $q = -4$ to 4. This research is able to expand the potential applications of perovskite microlasers in hybrid integrated photonic networks, as well as optical computing.

[1] State Key Laboratory on Tunable laser Technology, Ministry of Industry and Information Technology Key Lab of Micro-Nano Optoelectronic Information System, Shenzhen Graduate School, Harbin Institute of Technology, Shenzhen 518055, P.R. China. [2] Collaborative Innovation Center of Extreme Optics, Shanxi University, Taiyuan 030006 Shanxi, P.R. China. [3] National Key Laboratory of Science and Technology on Advanced Composites in Special Environments, Harbin Institute of Technology, Harbin 150080, P.R. China. [4]These authors contributed equally: Wenzhao Sun, Yilin Liu. ✉email: shumin.xiao@hit.edu.cn

An optical vortex is a light beam with helical phase rotating azimuthally around its optical axis, characterized by a phase singularity with vanishing intensity at the center[1–4]. The vortex nature makes the structured beams essential for optical manipulation and the developments of molecular machines. Recently, vortex beams with different topological charge $q$ are found to be mutually orthogonal and can therefore be multiplexed[5–7]. As a result, the generation of on-chip integrated orbital angular momentum (OAM) microlasers has been recognized as an effective approach to address the exponentially growing demand for worldwide network capacity[8–11]. In the past few years, great success has been achieved in on-chip integrated vortex microlasers. Highly directional laser emissions with well-defined topological charge have been experimentally achieved from microring with asymmetrical scatters and the chain of microdisks. In spite of the rapid breakthroughs, the current III–V semiconductor microlasers face the difficulty of being imbedded into modern Si or $Si_3N_4$ based photonic integrated circuits (PIC), significantly restricting their practical applications[12].

The solution-processed lead halide perovskites have the potential to tackle this problem[13–18]. Their intrinsic advantages such as long carrier diffusion length and low defect density have ensured the exceptional optical gain of lead halide perovskites[19–21]. Meanwhile, lead halide perovskites are compatible with many material types, including flexible substrates and metal electrodes[22–24]. This characteristic, associated with its high refractive index, allows a direct incorporation of a perovskite microlaser onto a $Si_3N_4$-based PIC. In the past 5 years, many types of perovskite lasers have been experimentally demonstrated, e.g., whispering gallery modes lasers, Fabry–Perot lasers, spasers, distributed feedback Bragg lasers, vertical cavity surface emitting lasers (VCSEL), and polariton lasers[4,19–22,24–41]. The corresponding applications are also extended to speckle-free illuminations, sensors, all-optical switch, and sub-diffraction limit imaging[4,39,42–44]. Despite of the continuous success, the current perovskite microlaser usually produces a stream of linearly polarized photons with uniform wavefront. The dependence on the complicated nanostructures and the inferior material properties have hindered the development of perovskite vortex microlaser, especially for the cases with topological charge $|q| > 1$. Herein, we employ the Archimedean spiral gratings and explore the perovskite VCSELs with well-defined topological charge. On-chip integrated vortex microlasers with $q = 0$, 1, 2, and up to 32 have been experimentally demonstrated.

## Results

### The preparation of MAPbBr$_3$ perovskite nanostructures.

The lead halide perovskite film was deposited with the vapor assisted spin-coating technique (see "Methods")[39]. The subsequent X-ray diffraction spectrum, energy dispersive spectroscopy (see Supplementary Figs. 2 and 3) are consistent with the previous report[6] and confirm the formation of MAPbBr$_3$ perovskite films. The thickness of perovskite film is controllable between 200 and 400 nm via the concentration of MAPbBr$_3$ precursor or the spin-coating speed. For simplicity, the thickness of MAPbBr$_3$ perovskite film is fixed at 300 nm with 20 nm variations in the following experiments. The atomic force microscope measurements show that the root mean square roughness of perovskite film is less than 6.15 nm (see Supplementary Fig. 1), good for optoelectronic devices.

The nanostructures were fabricated within the MAPbBr$_3$ perovskite films with a combined process of electron beam lithography and inductively coupled plasma etching (see "Methods"). In contrast to the previous reports[4,25,45], the etching gas Cl$_2$ is replaced with a mixture of H$_2$ and Ar. This change

brings two improvements. On one hand, the ion exchange process between Cl$^-$ and Br$^-$ has been eliminated. The corresponding photoluminescence measurement shows that the emission characteristics have been perfectly preserved during the etching process (see Supplementary Fig. 5). On the other hand, the etching rate can be well controlled to ~21 nm/s with a depth control accuracy of 20 nm (see Supplementary Fig. 6), which enables the shallow etching and will be essential for on-chip applications. The above information has been further confirmed with a perovskite grating. The period ($p$) and gap size ($g$) of the grating are 282 and 84 nm, respectively. The etching depth is controlled to around 200 nm. As shown in the Supplementary Fig. 9, single-mode laser emission has been simply obtained at 540 nm. This is consistent with the numerical design and confirms the precise control of our nanofabrication process.

### The Bullseye VCSELs.

While directional outputs and single-mode operation can be achieved with conventional gratings, such perovskite microlasers only produce the Gaussian beams, similar to the previous VCSELs[31–35]. To meet the requirements of biological imaging and optical manipulation, we study the lasing actions in two-dimensional grating, which is schematically shown in Fig. 1a. It is a circular Bragg grating on a 13 nm indium tin oxide (ITO)-coated glass substrate, also known as Bullseye grating. The period of Bullseye grating is $p = 282$ nm and the duty ratio is 0.7. The grating is also shallowly etched 200 nm and covered with another 300 nm PMMA film. An air defect is designed at the center of the bullseye. The resonances in the bullseye were calculated with three-dimensional numerical simulation (see "Methods"). As depicted in Fig. 1b, only one mode with relatively high Q factor was achieved in a wide spectral range. The corresponding field pattern (see Supplementary Figs. 9 and 10) shows that this mode is mainly confined within the perovskite waveguide. By controlling the size of air defect, a phase difference of $\pi$ can be introduced to the waves between gratings on both sides of the defect. In this case, the far-field vertical radiation along $\theta_{FF} = 0$ is destructively canceled (See Fig. 1c). Owing to the rotational symmetry of the bullseye, the far field radiation can thus be the desirable donut beam with high directionality.

Then the bullseye circular grating was fabricated experimentally. The top-view scanning electron microscope (SEM) image is depicted in Fig. 1d. The lattice size, duty ratio, defect size and the etching depth all follow the design well. The optical properties of the perovskite bullseye were characterized via optical pumping with a femtosecond laser (see "Methods"). At low pump power, only a broad photoluminescence peak centered at 536 nm can be seen (black line in Fig. 1e). With the increase of pump fluence, a sharp peak appears at the 545 nm (blueline in Fig. 1e). The intensity of the sharp peak increases dramatically with further increase of pump fluence and rapidly dominates the emission spectrum (brown line in Fig. 1e). Such a change can be more clearly seen in the log–log plot of the dependence of integrated output intensity on the pump fluence. As the dots shown in Fig. 1f, the slope of output intensity increases from ~0.98 to 3.13 and back to 1 with the increase of pump fluence. Simultaneously, the full width at the half maximum (FWHM) reduces from ~24 to 0.4 nm, consistent with the numerical calculation in Fig. 1. These experimental observations confirm the single-mode laser operation with a threshold of 19.58 μJ/cm$^2$ in the perovskite bullseye. The single mode operation is well preserved from the laser threshold to the gain saturation.

The far field patterns of the Bullseye microlaser was studied with the back focal plane imaging technique (see Supplementary Fig. 11). Figure 2a shows the beam profiles of bullseye lasers in

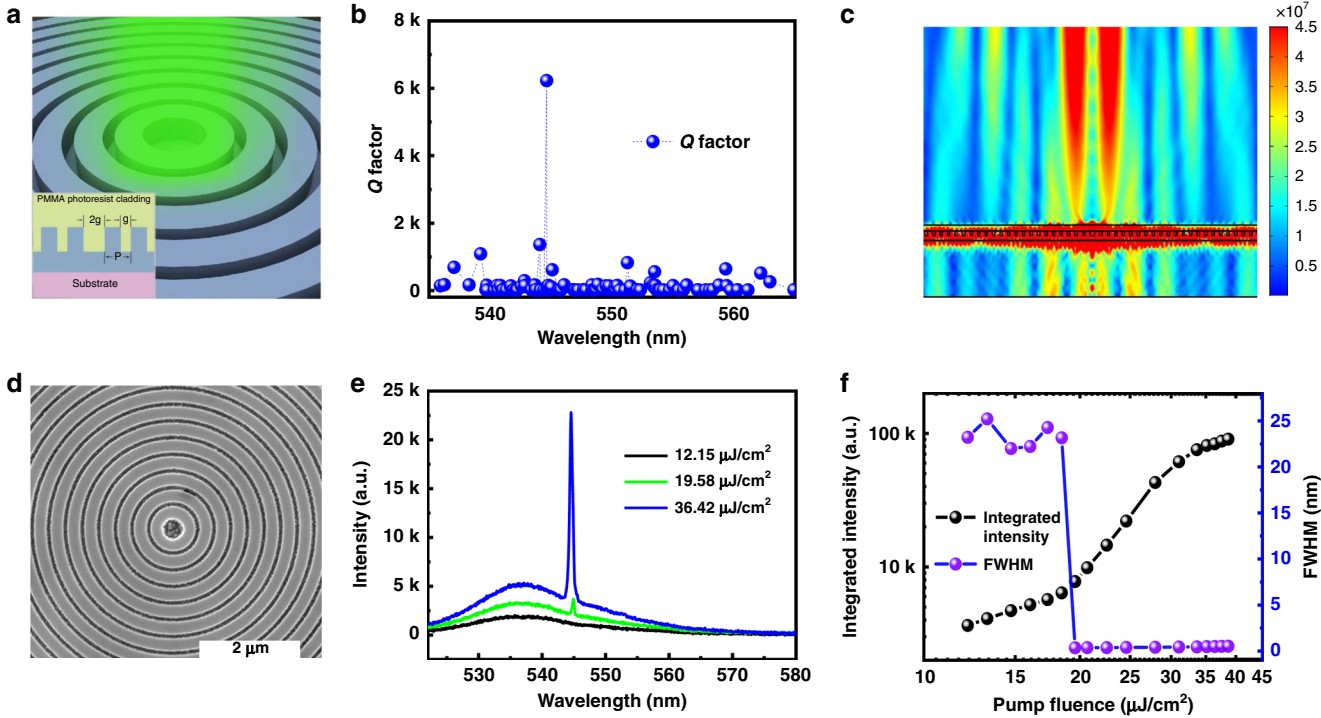

**Fig. 1 The Bullseye VCSELs. a** The schematic of the bullseye VCSEL. **b** The resonances in the perovskite bullseye. Here, the size parameters are $p = 282$ nm and $g = 84$ nm. **c** The mode profile and its far field emission of the high Q mode in (**b**). The divergence angle is around 3°. **d** The top-view SEM image of the perovskite bullseye nanostructures. **e** The evolution of emission spectrum with the increase of pump fluence. **f** The output intensity (black dotted line) and the FWHM (purple dotted line) as a function of pump fluence.

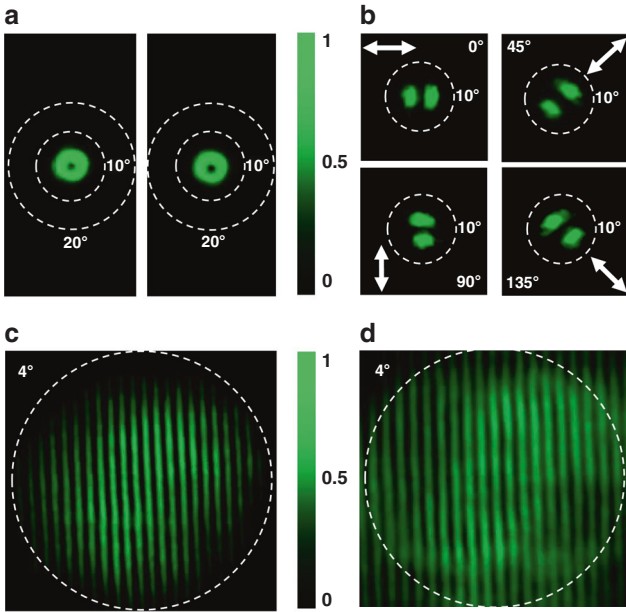

**Fig. 2 The far-field characteristics of perovskite bullseye VCSELs. a** The beam profiles of bullseye VCSEL recorded in the forward direction and backward direction. The white dashed circles represent 10° and 20° as an angular scale bar. **b** The polarization state of the donut shaped laser beam after a linear polarizer. **c, d** The self-interference patterns of the forward and backward donut, respectively. The white dashed circle corresponds to the angle of 4 degree. Here the pump fluence is 20.5 μJ/cm².

forward (left panel) and backward (right panel) directions, respectively. As expected, donut shaped laser beams can be clearly seen in both directions. The laser intensity at the center vanishes and the bright rings are around 3.5° from the normal direction (see Supplementary Fig. 13). These donut beams demonstrate vertically emitting characteristics of Bullseye microlasers very well. The polarization states of the Bullseye VCSELs were also characterized with a linear polarizer. As shown in Fig. 2b, the donut shape changes to two lobes and the direction of two lobes follows the linear polarizer, indicating the radial polarization of the Bullseye VCSELs. These unique characteristics make the bullseye VCSELs essential for some particular applications such as biological imaging and optical manipulation[46]. Figure 2c, d shows the self-interference patterns of Bullseye VCSELs. In both directions, only interference fringes have been observed. Therefore, the radially polarized donut from the perovskite VCSELs is a vector beam. The perovskite VCSELs also have very nice stability. No obvious reduction of the laser emission intensity has been observed after $3.6 \times 10^6$ continuous pump with a fluence of 37 μJ/cm² (see Supplementary Fig. 20 in the Supplementary information).

**The vortex microlasers with different topological charges.** To impose the OAM to the perovskite VCSELs, we have replaced the circular Bragg gratings with the Archimedean spiral grating. The inner and outer boundaries of the spiral grating are defined in polar coordinates following the equations $\rho(\theta) = p\left(0 + \frac{l\theta}{2\pi}\right)$ and $\rho(\theta) = p\left(f + \frac{l\theta}{2\pi}\right)$. Here, $p$, $f$ and $l$ are the lattice size, duty ratio, and the number of arms, respectively. The starting points of multiple arms are $\theta = 0$, $2\pi/l$, $(l-1)2\pi/l$, respectively. Compared with the Bullseye, the Archimedean spirals have very special features. Taking the Archimedean spiral with $l = 1$ as an example,

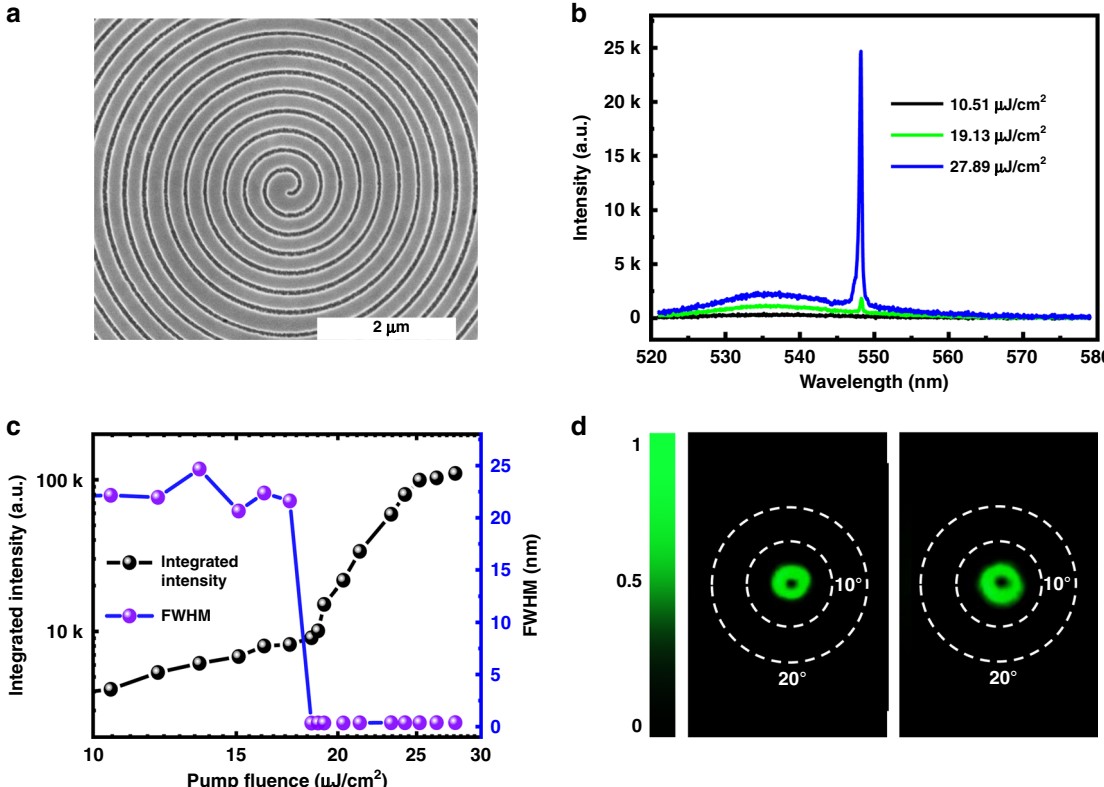

**Fig. 3 The laser characteristics of Archimedean spiral gratings. a** The top-view SEM image of the MAPbBr$_3$ perovskite Archimedean spiral grating. **b** The emission spectra of spiral grating at different pumping powers. **c** The dependence of output intensity (solid line) and FWHM (open squares) on the pump fluence. **d** The corresponding far field laser profiles in the forward direction and backward direction. The white dashed lines represent the angles of 10° and 20°, respectively.

it is a grating with fixed grating lattice size $p$ in the radial direction. In the azimuthal direction, the spiral shape can impose a $2\pi$ phase shift along a circle. Consequently, the Archimedean spiral gratings can endow a phase singularity to a resonant mode within the radial gratings and thus produce the vortex beams carrying the OAM.

With the above analysis, the Archimedean spiral gratings were experimentally fabricated within the 300 nm MAPbBr$_3$ perovskite film. Figure 3a shows the top-view SEM image of one Archimedean spiral grating. Due to the difference in thickness, here the grating with lattice size of $p = 288$ nm and duty ratio of 0.7 can match the gain spectral region of MAPbBr$_3$. The MAPbBr$_3$ spiral grating was optically excited to explore its laser performances. The results are summarized in Fig. 3b. With the increase of pump fluence, there is also a clear transition from a broad photoluminescence peak a single spike at 548 nm. The dependence of the integrated output on the pump fluence (see Fig. 3c) shows a clear "S" shaped curve. The slope changes from 1 to 4.31 and back to 1.13. This information, associated with the dramatic reduction of the FWHM at ~19.13 μJ/cm$^2$, confirms the onset of lasing action in the perovskite Archimedean spiral. Note that the Archimedean spiral grating also supports the single mode operation. The ration between the laser peak and the photoluminescence is more than 10 dB at the largest pump fluence (see Supplementary Fig. 21 in the Supplementary information).

With the above threshold excitation, highly directional laser emissions from the Archimedean spiral grating have been achieved in both of the forward and backward directions. Figure 3d shows the far field beam profiles. Similar to the Bullseye VCSELs, the Archimedean spiral laser emissions are perpendicular to the grating plane. They are also donut shapes

with vanishing centers. The divergent angles are around 3.7°, demonstrating the VCSELs characteristics well.

Figure 4a shows the experimentally recorded polarization states of the Archimedean spiral VCSEL. The donut laser beam becomes two lobes after passing a linear polarizer. The direction of two lobes follows the direction of linear polarizer, indicating the radial polarization with a singular point at the center. All of these observations are similar to the Bullseye VCSELs and confirm the successful generation of vector laser beams. The interesting phenomena happened in the self-interference experiments. Figure 4b shows the self-interference pattern of the forward laser beam. Different from the Bullseye VCSELs, there are a pair of inverted forks, which is a direct confirmation of the phase singularity within the Archimedean VCSELs. Figure 4c shows the self-interference pattern of the backward laser beam. There are also a pair of inverted folks with completely opposite directions to the forward ones. Therefore, we conclude that Archimedean spiral can produce two laser beams with OAM $q = -1$ and $+1$, respectively. Such a phenomenon is consistent with the geometry of the Archimedean spiral. The retrieved phase from the corresponding numerical calculation using Lumerical FDTD analysis shows a phase singularity at the center of the beam (see Fig. 4d). The phase shift along a circle surrounding the singularity is $2\pi$. As a result, vortex microlasers with reversed topological charge $q = \pm 1$ are obtained in two directions.

According to the equations, the arm number $l$ of the Archimedean spiral is controllable. With the increase of $l$ from 1 to 4, the Archimedean spiral starts to have 2-fold, 3-fold, and 4-fold rotational symmetries. The phase shift along a circle thus becomes $4\pi$, $6\pi$, and $8\pi$, respectively. As the lattice size and duty ratio are kept as the same, the lasing wavelength of the grating are

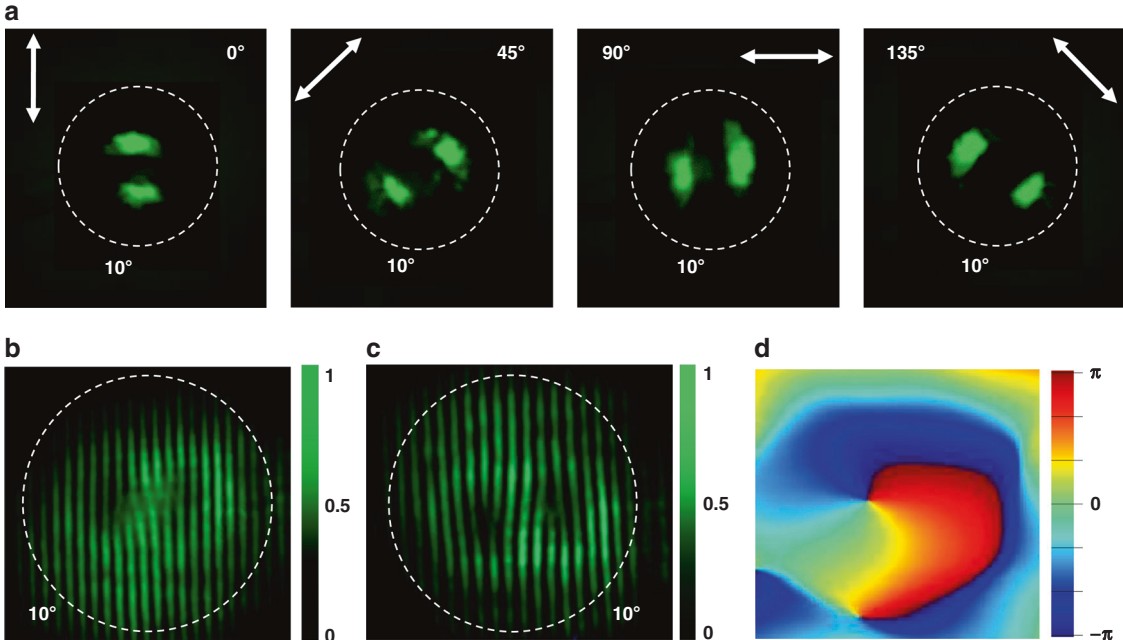

**Fig. 4 The far field characteristics of the Archimedean VCSELs. a** The polarization states of the laser beam. The divergence angle are marked with dashed lines. **b, c** The measured self-interference patterns of the laser beams in the forward direction and in the backward direction, respectively. Here, the pump fluence is 21 μJ/cm². **d** The numerically calculated phase profile of emission from the Archimedean spiral grating. The white dashed lines represent the angle of 10°.

also well preserved. As a result, the OAM of the laser emission from the Archimedean spiral gratings also becomes controllable, whereas the lasing wavelength can be pretty much preserved. We then fabricated a series of Archimedean spiral gratings with arm numbers of $l = 2, 3, 4$.

Figure 5a shows the top-view SEM images of these gratings, associated with the SEM images in Figs. 1c and 3a. The corresponding laser characteristics of the Archimedean spiral gratings have been studied under optical excitation. The corresponding laser spectra and thresholds can be seen in the Supplementary Figs. 16–18. The far field laser beam profiles from the perovskite spiral gratings are shown in Fig. 5b. With the increase of arm number from 0 to 4, the far field laser beams are all directional donut beams. The center of the donut gets larger and the divergent angles are 3.15°, 3.7°, 4.46°, 5.35°, 6.07°, consistent with the profile equation of OAM beam[40]. The corresponding self-interference patterns of the forward beam in Fig. 5c show 2–4 pairs of inverted forks when $l = 2, 3, 4$. These results, associated with the numerically calculated phase profiles in Fig. 5d, confirm that the topological number of OAM increases from 2 to 4 with the increase of arm number. These results are retrieved through Lumerical FDTD analysis after matching the simulated result to experiment. The self-interference patterns of the backward beams have also been measured. As shown in Supplementary Fig. 12, the directions of inverted forks are all opposite to the ones in Fig. 5c, demonstrating the topological numbers from −2 to −4 as well. Therefore, with the control of the bema number, the perovskite-based Bullseye and Archimedean spiral VCSELs have the capability of producing vortex laser beams with OAM $q = -4$ to 4.

It is important to note that the difference between Archimedean spiral and the bounded states in the continuum. While both of them can produce vortex laser emission. The topological charge is limited to small value due to the complicated nanostructures. In contrast, the topological charge of Archimedean spiral is determined by the arm number. With the increase

of arm number, perovskite VCSELs with topological charge $q = 8, 16, 32$ have also been experimentally demonstrated (see Supplementary Figs. 22–24 in the Supplementary information). Such a large range of topological charge makes perovskite VCSELs essential for high density mode division multiplexing.

## Discussion

In summary, we have experimentally fabricated a series of Bullseye grating and Archimedean spiral grating with different arm number of $l = 1, 2, 3$, and 4. With the precise control of nanofabrication, the lasing wavelengths are well preserved at around 540 nm, whereas the topological charge of the highly directional laser beams can be precisely tuned from $q = -4$ to $q = 4$. This is the first time that perovskite VCSELs can carry OAM with larges topological charges. The developed method can be expanded to larger topological charge by fabricating Archimedean spiral with more arms or spiral ring grating with more notches (see Supplementary Figs. 22–24 in the Supplementary information). We believe this research shall route a way of perovskite optoelectronic devices to on-chip optical network, biological sensing, quantum optics, and optical manipulations as well.

## Methods

**Numerical calculations**. The numerical calculation is based on a commercial finite element method package (COMSOL Multiphysics 5.1). Eigenfrequency analysis module is used to obtain the resonance mode diagrams. Owing to the rotational symmetry, only the cross section along the radial direction is calculated. The material refractive indices are measured by ellipsometer and shown in Supplementary Fig. 4 in the Supplementary information.

**The preparation of perovskite thin film**. The 1.4 mM/mL MAPbX₃ precursor was obtained by dissolving PbBr₂ and MABr (99.999%, Shanghai MaterWin New Materials Co.) with a 1:1 molar ratio in dimethyl sulfoxide for 6 h stirring (with density of 1.4 mM/mL). The 13 nm ITO coated substrate was hydrophilic treated with oxygen in a plasma cleaner (Femto) of 60 W power for 3 min. Then the MAPbBr₃ precursor solution was spin-coated onto the substrate for 90 s. At the twenty-third second, 70 μL chlorobenzene solution was casted onto the film to

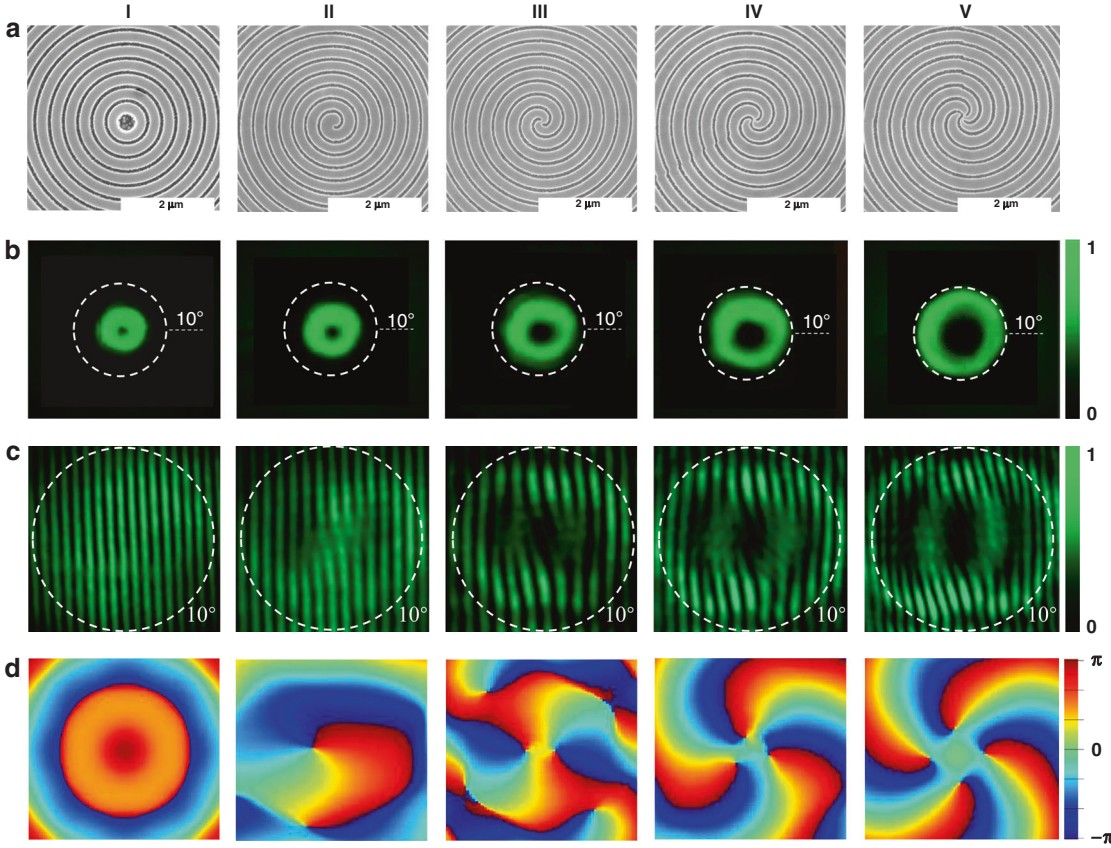

**Fig. 5 The MAPbBr$_3$ perovskite VCSELs with different topological charges. a** The top-view SEM images of the Archimedean spiral gratings with $l = 0$ (I), $l = 1$ (II), $l = 2$ (III), $l = 3$ (IV), and $l = 4$ (V). **b** The corresponding far field beam profiles of the perovskite VCSELs. **c** The corresponding self-interference patterns of emission beams in the forward direction. Here, the pump fluence is 21 µJ/cm$^2$. **d** The corresponding phase distribution simulation results. The white lines represent the angle of 10°.

form dense and uniform film. The thickness of perovskite film was controlled to 300 nm.

**Nanofabrication.** The MAPbBr$_3$ perovskite film was coated with 150 nm electron beam resist (ZEP520 A) and patterned by electron beam lithography (Raith Eline 150Plus). To avoid degrading the film quality, the acceleration voltage was controlled at 30 kV and the dose was around 70 µC/cm$^2$. The designed nanostructures were generated in the resist after developing in N50 for 50 s. Then the nanostructures were transferred into the perovskite film via the reactive ion etching (Oxford PlasmaPro 800Plus). After the etching process, the sample was quickly put into the glove box and spin-coated with 300 nm PMMA resist under negative pressure. The PMMA resist fills the gap regions and cover the whole sample very well.

**Laser characterization.** The sample was mounted on a three-dimensional translation stage under a home-made microscope, and excited by a frequency doubled laser (400 nm, using a BBO crystal) from a regenerative amplifier (repetition rate1 kHz, pulse width 100 fs, seeded by MaiTai, Spectra Physics). The pump light was focused to a 20 micron in diameter spot on the top surface of the sample through a 40× objective lens. The pump laser beam is roughly half of the perovskite grating (see Supplementary Fig. 15 in the Supplementary information). The emitted laser light was collected by the same objective lens and coupled to a spectrometer (Princeton Instruments, PIXIS UV-enhanced CCD) coupling spectrometer (Acton SpectroPro s2700) through a multimode fiber.

**Self-interference experiment.** The pump laser was the same as the above and the setup was shown in Supplementary Fig. 12. During the measurement, the laser emission was divided into two beams with equal intensities. A delay line was added to the optical path of one beam. By adjusting the delay line, the two optical paths had the same length and two beams thus met again. By tuning the bright ring of one beam overlap with the dark center of the other one, the self-interference patterns were produced.

**Reporting summary.** Further information on research design is available in the Nature Research Reporting Summary linked to this article.

## Data availability

All relevant data can be obtained from the corresponding author under reasonable request.

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

## Acknowledgements

This research is financially supported by National Key Research and Development Program of China under grant No. 2018YFB2200400, National Natural Science Foundation of China under grant Nos. 11974092, 61975041, 91850204, and 11934012, and National Natural Science Shenzhen Fundamental research projects under grant Nos. JCYJ20180507183532343, JCYJ20180507184613841, and JCYJ20180306172041577.

## Author contributions

S.X. conceived the concept and supervised the experiment. W.S. and Y.L. fabricated the samples. W.S., Y.L., G.Q., Y.F., W.D., and Y.W. performed the optical characterizations. W.S. did the numerical calculations. All the authors analyzed the data. S.X., Q.S., and J.H. wrote the paper.

## Competing interests

The authors declare no competing interests.
