## [Peer Review File · Nature Communications]

REVIEWER COMMENTS

Reviewer #1 (Remarks to the Author):

The manuscript by Q. Song et al presents an approach with experimental demonstration to perovskite vortex microlasers by introducing a nanograting into a perovskite film. By fabricating different Archimedean spiral gratings, the perovskite vortex microlasers could emit optical vortices with different topological charges. The output vortex mode is related to the specific grating, but cannot be manipulated dynamically. The technique and its supporting experimental results reported in the manuscript are convincing and sophisticated, could be of interest to the optics community.

I do not think, however, that this work is significant and novel enough to meet the criterion of Nature Communications. This is mainly due to the fact that it only provides a trivial improvement (i. e. changing the structures from nanowires to gratings) compared to the reported perovskite lasers (e. g. Gu Z, et al. *Advanced Optical Materials*, 4(3), 2015; Zhang Q, et al. *Nano Letters*, 14(10), 2014; Ref [43], etc.), without any real conceptual or practical advantages. For journal like Nature Communications, it pursues profound works which could impact the general science research community with fundamental mechanism novelty or striking application potentials, and these aspects are hardly seen in this case. Therefore, I believe that this manuscript could be more appropriate for publication in a specialized journal in optical materials or photonics areas.

Besides, there are several typos in the References, the authors may need to check their manuscript more carefully.

Reviewer #2 (Remarks to the Author):

NCOMMS-20-15607

The manuscript from W. Sun et al. describes a perovskite based microlaser scheme allowing the generation of highly directional vortex beams. The large refractive index of lead halide perovskites, combined with a large optical gain, enables the fabrication of microlasers where the feedback element (grating) is also the active material. Using five different microstructures the authors trigger lasing in radially polarized modes with orbital angular momentum (OAM) charges $|q|=0,\dots,4$. Due to the change of the handedness under mirror symmetry, the emission from the top and bottom of the structures presents opposite OAM. As the rotational symmetry of the structure is explicitly broken in spiral grating structures, the emission OAM is imposed at the fabrication step.

The main claim of the article, namely the generation of vortex beams with $|q|>1$ in a perovskite-based microstructure, is well documented by the experimental evidence presented in the manuscript. I particularly appreciated the rich supplementary materials detailing, among others, an improved etching technique. Although this nice work constitutes the first demonstration of perovskite-based microlasers emitting higher order vortex beams, potentially compatible with photonic integrated circuits, a similar scheme was implemented in ref. 10 using organic materials. In my opinion, it thus constitutes a relevant technological but not conceptual advance for the field.

However, a more quantitative analysis of the microlaser emission properties would extend the scope of the paper and its relevance for the community. This would provide a stronger argument to recommend publication.

I append some remarks and questions for the authors below:

1) In all the measured far-field emission patterns in the main text there is no scale: although a calibration is presented in the supplementary materials (Fig. S11), a scale is necessary to compare the far-field and self-interference images in Fig. 2,4,5. At what power density were the images in Fig. 2 and 4 taken? The authors should also add colour gradient indicators allowing the reader to deduce the relative intensity of the different features in the image. For instance: which is the degree of radial polarization of the beam in Fig. 2 and 4? Are the intensities of all the spatial patterns normalized? In Fig. 2a, 5b and 5c I have the impression that the intensity is saturated.

2) The measured full-width at half maximum of the emission peak above threshold (1.5nm) does not appear to be limited by the resolution of the spectrometer, nor by the expected cavity mode linewidth (0.5 nm), nor by the Fourier-limited envelope of the 100fs pulsed excitation (5.3 nm @ 400nm). Can the authors explain what is limiting the spectral width of the microlaser emission peak?

3) In the interferograms presented in Fig. 5 the phase singularities (pitchfork bifurcations) scatter across all the far-field. I would expect them to concentrate about the optical axis of the two beams producing the interferogram. When comparing Fig. S9 and S10, it seems that the lens used to image the objective back-focal plane in Fig. S9 has been removed for the interferometric measurements. If this is the case, which is the reason for such choice? This might also explain the sparseness of the pitchfork bifurcations in the interferograms. Can the authors provide the phase-maps associated to the fringe patterns to help the reader visualize the optical vortices?

4) What is limiting the purity of the lasing mode radial polarization? What is the degree of linear polarization of the emission as a function of the angle of the polarizer axis?

5) (Possibly related to points 2 and 4) In all emission spectra above threshold [Fig. 1e, Fig. 3b and Fig. S13a-S15a] the lasing peak presents a shoulder on the longer wavelengths side. Do the authors have an explanation for this feature? I wonder if it is possible that the bullseye grating does not present a perfect rotational symmetry, yielding two preferential polarization axes and lifting the degeneracy between the transverse modes. Then, the laser emission would present a double peaked spectrum and an elliptically polarized spatial pattern. If this is the case, the authors should remove the claim of single-mode operation.

6) In the conclusion, the authors indicate the possibility to extend this method to larger topological charges. Isn't it there any constraint related to the number of arms in the spiral grating and the smallest feature which etched? Up to which OAM value they expect their scheme to be realistically extended?

Minor remarks:

7) Is the emission peak energy constant as a function of the input power density? Does the laser keep to be single mode above the saturation threshold?

8) Comparing the spectra and IP curves in Fig.2-(e,f) or Fig.3-(b,c) I have the impression that the intensities reported in the IP plots correspond to either the peak intensity at the lasing mode wavelength ($\lambda_0 \sim 545$ nm), or to the integrated intensity in a few nm bandwidth about λ_0 . If this is the case, the IP curves should be corrected with the integrated intensity over all the spectrum. Anyway, the label of the y axis in the IP curve plots should be changed to "Integrated intensity".

9) Line 143: which thickness difference are the authors referring to? I believed the thickness of the perovskite layer was roughly constant among the different devices as stated in line 61.

10) In most of the far-field emission profiles there is a spurious dark feature near the left side, can the authors explain what it is related to?

11) Line 350: I guess the authors intend 20 (W) power.

I found some typos listed below. Other suggestions are within brackets:

Main text:

Line 90: far field emission (pattern) can thus be [...]. Line 154: Shaped, [...] vanishing (intensity on the optical axis). Line 163: laser beam (presents) two lobes. Line 170: forks. Line 193: beams. Line 199: beam. Line 212: larger. Line 345: ellipsometry.

Supplementary materials:

Line 59: degrees. Line 93-94: onset of (lasing in perovskite grating microstructures). Line 125: achieve, image. Line 130: self interference, essential. Line 136: produce. Line 146: degrees, divergence. Line 148: divergence. Line 153: charges. Line 179: there (are) no forks. Line 181: [...], (well consistent with the theoretical expectations).

Reviewer #3 (Remarks to the Author):

The work "Lead halide perovskite vortex microlasers" submitted to Nature Communication reports on the experimental demonstration of the perovskite vortex microlasers in the form of halide perovskite gratings. The experimental results are supported by numerical modeling. The work is well-organized, very timely, and important for the field of perovskite-based photonics. Thus, I believe it deserves to be published after addressing some technical comments:

- 1) Fig.3c: I am not sure that "power density" is the correct name for the value corresponding to pulse energy per cm^2 , because "power" deals with temporal characteristics of the laser pulse, which is not the case here. Usually, it is called "fluence".
- 2) Fig.3b: the authors show spectra with single-mode lasing below a fluence of 40 uJ/cm^2 . However, it would be useful to show the spectra at fluences higher than 40 uJ/cm^2 . Is there any dramatic change to the multimode regime of lasing?
- 3) It is important to give the fluence values for Figs. 3d, 4, and 5 as well as give corresponding PL spectra. Indeed, there should be information on the ratio between spontaneous PL background and intensity of lasing mode.
- 4) The etching procedure of the perovskites might affect the PL quantum yield. Thus, it is important to show any comparison of PL properties before and after the etching.
- 5) I encourage to explicitly show the image with pumping beam distribution on the gratings to understand how many periods are involved in the process.
- 6) The authors wrote "the material refractive indices are measured by ellipsometer". However, I did not find the corresponding plot with these values.
- 7) The text quality should be improved. I found a number of typos like "to each other. and two emission".
- 8) In Ref.[4] the authors also observed vortex beam generation from nanopatterned perovskite. Can they compare the current results with their previous work to show the advantages and differences more clearly?

Reply to Reviewer #1

First of all, we would like to thank the reviewer for the very careful review and the recognition of our observations. We have carefully read the reviewer's report and replied in details below.

Comment-1: The manuscript by Q. Song et al presents an approach with experimental demonstration to perovskite vortex microlasers by introducing a nanograting into a perovskite film. By fabricating different Archimedean spiral gratings, the perovskite vortex microlasers could emit optical vortices with different topological charges. The output vortex mode is related to the specific grating, but cannot be manipulated dynamically. The technique and its supporting experimental results reported in the manuscript are convincing and sophisticated, could be of interest to the optics community.

I do not think, however, that this work is significant and novel enough to meet the criterion of Nature Communications. This is mainly due to the fact that it only provides a trivial improvement (i. e. changing the structures from nanowires to gratings) compared to the reported perovskite lasers (e. g. Gu Z, et al. *Advanced Optical Materials*, 4(3), 2015; Zhang Q, et al. *Nano Letters*, 14(10), 2014; Ref [43], etc.), without any real conceptual or practical advantages. For journal like Nature Communications, it pursues profound works which could impact the general science research community with fundamental mechanism novelty or striking application potentials, and these aspects are hardly seen in this case. Therefore, I believe that this manuscript could be more appropriate for publication in a specialized journal in optical materials or photonics areas.

Our response: We thank the reviewer for the recognition of our results. We are sorry that we haven't clearly stated our main contributions in our previous version. We have to note here that this research is not a simple and trivial improvement from the previous researches. All of the previously reported perovskite lasers are linearly polarized and have Gaussian shaped laser beams. Such kind of microlasers can be on-chip integrated and of course have important applications. However, they are limited in high density mode division multiplexing. In contrast, the vortex beams with orbital angular momentum (OAM) are highly desirable in such applications. In principle, vortex beams with different topological charge q are orthogonal and can therefore be multiplexed. As a result, the vortex lasers have very promising potential to expand the capacity of optical communications and have been intensively studied.

Very recently, the on-chip integrated vortex microlasers have attracted considerable research attention. For such application, the solution processed lead halide perovskites are very promising. This material is easy to achieve and compatible with the current photonic chips. Meanwhile, it has exceptional gain coefficient and higher refractive index than Si_3N_4 . These

characteristics make lead halide perovskites to possibly tackle the long-standing problem of on-chip integrated coherent light sources. Therefore, the combination of vortex microlasers and lead halide perovskites are highly desirable and essential.

In this research, we experimentally demonstrate the perovskite vortex microlasers with highly directional outputs and well-controlled topological charges. We have fabricated high quality perovskite gratings in perovskite film with a precise control of the grating size and etching depth. Consequently, vertical cavity surface emitting lasers (VCSELs) with divergent angles of 3° have been successfully achieved. By simply controlling the Archimedean spiral, the wavefront of the perovskite VCSELs can be switched to be helical with topological charges of $q = \pm 1, \pm 2, \pm 3, \text{ and } \pm 4$, respectively. By further increasing the arm number, we have also realized the emission with topological charge of 8, 16, and 32. The topological charge of 32 is the current record value for on-chip integrated vortex microlasers. Therefore, all of these observations are intrinsically different from the previous reports, e.g. *Advanced Optical Materials*, 4, 472-479 (2015), *Nano Letters*, 14(10), 5995-6001 (2014), and Ref. [43].

In the revised manuscript, we have added the corresponding information in Para-1, Page-1, Para-1, Page-2, Para-2, Page-2, and Para-2, Page-10.

“However, most perovskite microlasers are linearly polarized with uniform wavefront. The structured laser beams carrying orbital angular momentum have rarely been studied and the applications of perovskites in next-generation optical communications are thus hindered.”

“Recently, vortex beams with different topological charge q are found to be mutually orthogonal and can therefore be multiplexed. [5-7] As a result, the generation of on-chip integrated OAM microlasers has been recognized as an effective approach to address the exponentially growing demand for worldwide network capacity. [8-11]”

“Despite of the continuous success, the current perovskite microlaser usually produces a stream of linearly polarized photons with uniform wavefront. The dependence on the complicated nanostructures and the inferior material properties have hindered the development of perovskite vortex microlaser, especially for the cases with topological charge $|q| > 1$. Herein we employ the Archimedean spiral gratings and explore the perovskite VCSELs with well-defined topological charge. On-chip integrated vortex microlasers with $q = 0, 1, 2$, and up to 32 have been experimentally demonstrated.”

“It is important to note that the difference between Archimedean spiral and the bounded states in the continuum (BICs). While both of them can produce vortex laser emission. The topological charge is limited to small value due to the complicated nanostructures. In contrast, the topological charge of Archimedean spiral is determined by the arm number. With the increase of arm number, perovskite VCSELs with topological charge $q = 8, 16, 32$ have also

been experimentally demonstrated (see Fig. S22 to Fig. S24 in the supplementary information). Such a large range of topological charge makes perovskite VCSELs essential for high density mode division multiplexing.”

Comment-2: Besides, there are several typos in the References, the authors may need to check their manuscript more carefully.

Our response: We thank the reviewer for the very careful review. Following the reviewer’s suggestion, we have carefully checked the references and removed several typos.

Reply to Reviewer #2

We thank the reviewer for the very careful review and valuable suggestions. We particularly appreciate the comments on the linewidth, the polarization purity, and the shoulder in the laser spectra. These comments let us realize that the sample quality is not perfect. Based on the reviewer's suggestions, we have improved our nanofabrication technique and made the following main revisions.

1. New samples have been fabricated and measured. With these new samples, single-mode operation has been achieved and the laser linewidth is improved to ~ 0.4 nm. The purity of polarization is also subsequently improved. All the corresponding figures in main text and SI are updated.
2. The topological charge number has been increased from 4 to 32. This is a significant step for the on-chip integrated vortex microlaser.
3. Numerical calculation and theoretical analysis have also been added.

In the revised manuscript, all the comments have been addressed accordingly and the quality of our research is significantly improved. The details can be seen below.

Comment-1: In all the measured far-field emission patterns in the main text there is no scale: although a calibration is presented in the supplementary materials (Fig. S11), a scale is necessary to compare the far-field and self-interference images in Fig. 2, 4, 5. At what power density were the images in Fig. 2 and 4 taken? The authors should also add colour gradient indicators allowing the reader to deduce the relative intensity of the different features in the image. For instance: which is the degree of radial polarization of the beam in Fig. 2 and 4? Are the intensities of all the spatial patterns normalized? In Fig. 2a, 5b and 5c I have the impression that the intensity is saturated.

Our response: We thank the reviewer for the very careful review and valuable comment. Based on the reviewer's comment, we have replotted the figures. The results are shown in Fig. R1-Fig. R3 below.

1. The divergent angle scale bar, the color bar, and the degree of polarization have been added in these figures.
2. The corresponding pumping densities are mentioned in the figure captions. The intensity in all the figures are normalized and the saturation problem has also been eliminated.
3. The degree of polarization has also been added in the figure.

Figure R1 (Fig. 2 in the main text): The far-field characteristics of perovskite bullseye VCSELs. (a) The beam profiles of bullseye VCSEL recorded in the forward direction and backward direction. The white dashed circles represent 10 degree and 20 degree as an angle scale bar. (b) The polarization state of the donut shaped laser beam after a linear polarizer. (c) and (d) are the self-interference patterns of the forward and backward donut, respectively. The white dashed line corresponds to the angle of 4 degree. Here the pump fluence is $20.5 \mu\text{J}/\text{cm}^2$.

Figure R2 (Fig. 4 in the main text): The far field characteristics of the Archimedean VCSELs. (a) The polarization states of the laser beam. The divergence angles are marked with dashed lines. (b) and (c) are the simulated self-interference patterns of the laser beams in the forward direction and in the backward direction, respectively. Here the pump fluence is $21 \mu\text{J}/\text{cm}^2$. (d) The numerically calculated phase profile of emission from the Archimedean spiral grating. The white dashed lines represent the angle of 10 degree.

Figure R3 (Fig. 5 in the main text): The MAPbBr₃ perovskite VCSELs with different topological charges. (a) The top-view SEM images of the Archimedean spiral gratings with $l = 0$ (I), $l = 1$ (II), $l = 2$ (III), $l = 3$ (IV), and $l = 4$ (V). (b) The corresponding far field beam profiles of the perovskite VCSELs. (c) The corresponding self-interference patterns of emission beams in the forward direction. Here the pump fluence is 21 \$\mu\text{J}/\text{cm}^2\$. (d) The corresponding phase distribution simulation results. The white lines represent the angle of 10 degree.

Comment-2: The measured full-width at half maximum of the emission peak above threshold (1.5nm) does not appear to be limited by the resolution of the spectrometer, nor by the expected cavity mode linewidth (0.5 nm), nor by the Fourier-limited envelope of the 100fs pulsed excitation (5.3 nm @ 400nm). Can the authors explain what is limiting the spectral width of the microlaser emission peak?

Our response: We thank the reviewer for this valuable comment. In our VCSEL, the Q factor of VCSEL should be mainly determined by the imperfect rotational symmetry and the radiation loss. The Q factor related the radiation loss is calculated shown in Fig. 1b in the main text. The corresponding linewidth is about 0.43 nm. In this sense, it is straightforward to know that the main loss comes from the imperfect rotational symmetry. This is also consistent with another important comment raised by the reviewer that a shoulder appears at higher pumping fluence.

To check this assumption, we have refabricated the samples by compensating the fabrication deformation in GDS file. The corresponding laser spectra are shown in Fig. R4 below. The single mode laser is well kept from laser threshold to gain saturation. No shoulder can be observed even the pump fluence is above saturation point. As a result, we know that the rotational symmetry has been significantly improved. Simultaneously, the laser linewidth at the transparent threshold has also been measured. It is reduced to ~ 0.4 nm, consistent with the numerical simulation very well. The slight difference should come from the resolution of spectrometer (0.1 nm) and the influence of material gain. Then we can confirm that the main loss in our previous version comes from the imperfect rotational symmetry.

In the revised manuscript, we have replaced the Fig. 1(e) and the description in Para-1, Page-5. “Simultaneously, the full width at the half maximum (FWHM) reduces from \$\sim 24\$ nm to 0.4 nm, consistent with the numerical calculation in Fig. 1.”

Figure R4 (Fig. 1(e) in the main text). The single mode laser spectrum of $q=0$ at different pumping fluence. With the increase of pumping density from threshold to gain saturation, the VCSEL keeps single-mode operation very well.

Comment-3: In the interferograms presented in Fig. 5 the phase singularities (pitchfork bifurcations) scatter across all the far-field. I would expect them to concentrate about the optical axis of the two beams producing the interferogram. When comparing Fig. S9 and S10, it seems that the lens used to image the objective back-focal plane in Fig. S9 has been removed for the interferometric measurements. If this is the case, which is the reason for such choice? This might also explain the sparseness of the pitchfork bifurcations in the interferograms. Can the authors provide the phase-maps associated to the fringe patterns to help the reader visualize the optical vortices?

Our response: We thank the reviewer for this valuable comment. We have checked the

optical setup and Fig. S9 and S10. There are some mistakes in the plots and they have been fixed in the revised manuscript.

The reviewer is absolutely right that the bifurcation of pitchfork comes from the optical setup. It is caused by the wedged beam splitter in previous setup. The wedged beam splitter introduces additional phase shifts and induces the bifurcation. In the revised manuscript, we have fixed the optical setup and re-measured the self-interference patterns. As depicted in Fig. R5 below, the bifurcation of pitchfork has been completely removed and the self-interference pattern is much better now.

Figure R5 (Fig. 5(c) in the main text). The self-interference pattern of VCSELs with topological charge of $q = 0, 1, 2, 3,$ and 4 .

In addition to the self-interference patterns, the phase profiles of the emissions from Archimedean spirals have also been calculated. All the results are shown in **Fig. R6** below. With the increase of arm number of Archimedean spirals, it is clear to see that the number of phase singularity increases from 0 to 4, consistent with the self-interference pattern as well.

Figure R6 (Fig. 5(d) in the main text). The phase distribution diagrams of the emissions from bullseye grating and Archimedean spiral grating with arm number of 1-4.

In the revised manuscript, we have added the above experimental results and numerical calculations in **Fig. 4** and **Fig. 5** in the main text. The corresponding experimental description has also been added in Para-1, Page-8 and Para-1, Page-10 in the main text.

“The corresponding numerical calculation shows a phase singularity at the center of the beam (see Fig. 4(d)). The phase shift along a circle surrounding the singularity is 2π . As a result, vortex microlasers with reversed topological charge $q = \pm 1$ are obtained in two directions.”

“These results, associated with the numerically calculated phase profiles in Fig. 5(d), confirm that the topological number of OAM increases from 2 to 4 with the increase of arm number.”

Comment-4: What is limiting the purity of the lasing mode radial polarization? What is the degree of linear polarization of the emission as a function of the angle of the polarizer axis?

Our response: We thank the reviewer for this valuable comment. We agree with the reviewer that our previous results are somehow strange. The polarization degree is not perfect. When the radially polarized donut beams pass through a linear polarizer, they look more like modulated rings instead of two lobes. This effect is also caused by the imperfect rotational symmetry of the nanograting. By compensating the imperfection, the new linearly polarized lobes are measured and shown in **Fig. R7**. The intensity distributions behind the linear polarizer are now very close to the theoretical predictions. We have also checked the degree of the linear polarization as a function of the angle of polarizer axis. In the new samples, we find that the linear polarization almost parallel to the polarizer axis with a deviation less than 5° . In the revised manuscript, we have replaced **Fig. 2(b)** with the new data.

Figure R7 (Fig. 2(b) in the main text). The radial polarization of VCSEL with $q = 0$. After passing a linear polarizer, the donut beam turns into two separated lobes along the direction of linear polarizer.

Comment-5: (Possibly related to points 2 and 4) In all emission spectra above threshold [Fig. 1e, Fig. 3b and Fig. S13a-S15a] the lasing peak presents a shoulder on the longer wavelengths side. Do the authors have an explanation for this feature? I wonder if it is possible that the bullseye grating does not present a perfect rotational symmetry, yielding two preferential polarization axes and lifting the degeneracy between the transverse modes. Then, the laser emission would present a double peaked spectrum and an elliptically polarized spatial pattern. If this is the case, the authors should remove the claim of single-mode operation.

Our response: We really appreciate the reviewer's very careful review and valuable suggestion. Following the reviewer's suggestion, we have checked our sample carefully. We indeed found the imperfect rotational symmetry in the samples that were caused by the

nanofabrication deviations. Then we compensated the deviations in GDS files and re-fabricated all the samples. The results are shown in Figure R8-R12 below. After removing the imperfection, all the samples show very nice single mode operation from laser threshold to gain saturation. No shoulders can be observed even at the highest pump fluence. The reviewer is absolutely right that the imperfect rotational symmetry also relates to comment 2 and comment 4. By removing the imperfection, the polarization two lobes become very clear and the linewidth is reduced to ~ 0.4 nm.

In the revised manuscript, we have replaced all the laser spectra and the LI curves in the main text and in the supplementary information. According to our current experimental results, we keep the claim of single mode operation in our paper.

Figure R8 (Fig. 1(e), (f) in the main text). (a) The laser spectra at different pumping fluence from the Bullseye grating. (b) The integrated output intensities (black dots) and the linewidth (purple dots) as a function of pump fluence.

Figure R9 (Fig. 2(b), (c) in the main text). (a) The laser spectra at different pumping fluence from the Archimedean grating with arm number $l = 1$. (b) The integrated output intensities (black dots) and the linewidth (purple dots) as a function of pump fluence.

Figure R10 (Fig. S16 in the Supplementary Information). (a) The laser spectra at different pumping fluence from the Archimedean grating with arm number $l = 2$. (b) The integrated output intensities (black dots) and the linewidth (purple dots) as a function of pump fluence.

Figure R11 (Fig. S17 in the Supplementary Information). (a) The laser spectra at different pumping fluence from the Archimedean grating with arm number $l = 3$. (b) The integrated output intensities (black dots) and the linewidth (purple dots) as a function of pump fluence.

Figure R12 (Fig. S18 in the Supplementary Information). (a) The laser spectra at different pumping fluence from the Archimedean grating with arm number $l = 4$. (b) The integrated output intensities (black dots) and the linewidth (purple dots) as a function of pump fluence.

In the revised manuscript, we have replaced all the emission spectra in the main text and in the supplementary information. The corresponding single mode operation information has been emphasized in Para-1, Page-5 and Para-1, Page-7 in the main text.

“The single mode operation is well preserved from the laser threshold to the gain saturation.”

“Note that the Archimedean spiral grating also supports the single mode operation. The ration between the laser peak and the photoluminescence is more than 10 dB at the largest pump fluence (see Fig. S21 in the supplementary information).”

Comment-6: In the conclusion, the authors indicate the possibility to extend this method to larger topological charges. Isn't it there any constraint related to the number of arms in the spiral grating and the smallest feature which etched? Up to which OAM value they expect their scheme to be realistically extended?

Our response: We thank the reviewer for this valuable comment. This comment encourages us to approach the limit of our scheme. Following the reviewer's suggestion, we have fabricated new samples and increased the arm number. The experimental results are summarized in **Figure R12** below. By increasing the arm numbers of Archimedean spiral to $l = 8, 16, 32$ (see SEM images), perovskite microlasers have been confirmed with their threshold behaviors and dramatic change in linewidth. Similar to the perovskite based Archimedean spirals with small arm numbers, the perovskite microlasers also keep the single mode operation (column-II) and the donut beam profiles (column-IV). The beams also change from donut to two lobes after passing through a linear polarizer (see **Figure R13**). The self-interference patterns of the laser beams have also been studied and shown in **Figure R14**, where 8, 16, and 32 pairs of inverted forks can be clearly seen. Therefore, we can confirm that our scheme can at least support the vortex beam with topological charge of 32.

Figure R13 (Fig. S22 in the Supplementary Information). The vortex laser emissions from Archimedean spirals with large arm number of $l = 8$ (a), 16 (b), and 32 (c). Columns-I, II, III, and IV show the top-view SEM images, the laser spectra at different pump fluence, threshold behaviors, and the back focal plane images of the vortex lasers.

The experimental results also show the limitations of our scheme. The column-IV of Figure R12 shows the back focal plane beam profiles of the perovskite lasers. With the increase of arm number, we can see that the divergent angle keeps increasing. For the case of $l = 32$, the divergent angle is almost 25 degree, which is too large for a lot of practical applications. This will be one limitation of our scheme. Another limitation comes from the nanofabrication. As depicted in column-I of Figure R12, the Archimedean spiral remains perfect with arm number of $l = 8$. However, for the case of $l = 16$ and 32, there are obvious defect are formed at the center. This will also affect the device performances such as laser threshold and beam uniformity etc. Such defect will be more dramatic at larger arm numbers. Based on these results, we would like to set the limitation below topological charge of $q = 32$. We should note here that the topological charge of $q = 32$ is already a record for the on-chip integrated vortex microlasers and should be useful for many practical applications.

Figure R14 (Fig. S23 in the Supplementary Information). The polarization states of the donut beams from Archimedean spirals with arm number of $l = 8$ (a), 16 (b), and 32 (c). From left column to the right column, the polarization axis is 0° , 45° , 90° , and 135° , respectively.

Figure R15 (Fig. S24 in the Supplementary Information). The self-coherent interference patterns of Archimedean spirals with arm number of $l = 8$ (a), 16 (b), and 32 (c).

In the revised manuscript, we have added the corresponding information in Para-2, Page-10 of the main text and Section-7 of the Supplementary Information.

“It is important to note that the difference between Archimedean spiral and the bounded states in the continuum (BICs). While both of them can produce vortex laser emission. The topological charge is limited to small value due to the complicated nanostructures. In contrast,

the topological charge of Archimedean spiral is determined by the arm number. With the increase of arm number, perovskite VCSELs with topological charge \$q = 8, 16, 32\$ have also been experimentally demonstrated (see Fig. S22 to Fig. S24 in the supplementary information). Such a large range of topological charge makes perovskite VCSELs essential for high density mode division multiplexing.”

Comment-7: Is the emission peak energy constant as a function of the input power density? Does the laser keep to be single mode above the saturation threshold?

Our response: We thank the reviewer for these two valuable comments. In our previous samples, the imperfect rotational symmetry increased the laser linewidths and introduced the shoulders. In our new samples, by removing the imperfection in rotational symmetry, as shown in all the above figures, the single mode operation is maintained even at the saturation threshold and higher pump fluence.

Figure R16 (Fig. S20 in the Supplementary Information). The stability of our perovskite microlasers.

The emission peak energy is constant as a function of the input power density. Taking the sample with $q=0$ as an example, we fixed the pump fluence and measured the output intensity as a function of laser shots. As shown in Figure R15, the laser intensity is only slightly reduced after they sample was pumped with 3.6×10^6 laser shots, clearly demonstrating the stability of our microlasers. The corresponding discussion has been added in **Para-1, Page-6** in the main text and the related results are shown in Section 6 in Supplementary Information.

“The perovskite VCSELs also have very nice stability. No obvious reduction has been

observed after \$3.6 \times 10^6\$ continuous pump with a fluence of \$37 \mu\text{J}/\text{cm}^2\$ (see Fig. S20 in the supplementary information).”

Comment-8: Comparing the spectra and IP curves in Fig.2-(e,f) or Fig.3-(b,c) I have the impression that the intensities reported in the IP plots correspond to either the peak intensity at the lasing mode wavelength ($\lambda_0 \sim 545 \text{ nm}$), or to the integrated intensity in a few nm bandwidth about λ_0 . If this is the case, the IP curves should be corrected with the integrated intensity over all the spectrum. Anyway, the label of the y axis in the IP curve plots should be changed to “Integrated intensity”.

Our response: We thank the reviewer for the careful review. The reviewer is correct that the IP curves are the integrated intensity of the spectra. The difference in value is caused by the integration time during the experiment. Following the reviewer’s suggestion, we have replaced the label of y axis in the IP curve to “**Integrated intensity (a.u.)**”.

Comment-9: Line 143: which thickness difference are the authors referring to? I believed the thickness of the perovskite layer was roughly constant among the different devices as stated in line 61.

Our response: We thank the reviewer for the very careful review. The reviewer is absolutely right that the thickness of perovskite layer was controlled at 300 nm during the experiment. However, the real experiments cannot be this accurate. There is always around $\pm 10 \text{ nm}$ variation in thickness. As a result, the lattice needs to change slightly to fit the same lasing wavelength. To avoid confusion, we have changed the statement in **Para-1, Page-3** of the main text. “For simplicity, the thickness of MAPbBr₃ perovskite film is fixed around 300 nm with 20 nm variations in the following experiments.”

Comment-10: In most of the far-field emission profiles there is a spurious dark feature near the left side, can the authors explain what it is related to?

Our response: We thank the reviewer for this very careful review. The reviewer is correct that there are always several spurious dark features in the far field patterns. These features are caused by the damaged regions of our CCD cameras. There are several regions on the CCD are damaged. During the revised experiment, we have to tune the samples slightly away from such regions. The dark feature has been eliminated.

Comment-11: Line 350: I guess the authors intend 20 (W) power.

Our response: We thank the reviewer for the very careful review. We are sorry that the 20% power in this sentence is very confusing. This is a meter number of the system and has been replaced to the exact power in Methods of the revised manuscript. “The 13 nm ITO coated substrate was hydrophilic treated with oxygen in a plasma cleaner (Diener Electronic, Femto

BMS) of 60W for 3 minutes”

Reply to Reviewer #3 :

We thank the reviewer for the careful review and valuable suggestions. Based on the reviewer's comments, we have carefully revised our manuscript and all the comments have been addressed accordingly. The details of revision can be seen below.

Comment-1: Fig.3c: I am not sure that “power density” is the correct name for the value corresponding to pulse energy per cm^2 , because “power” deals with temporal characteristics of the laser pulse, which is not the case here. Usually, it is called “fluence”.

Our response: We thank the reviewer for the very careful review. The reviewer is correct that the power density is not accurate. The word “power” is a typo of “pump” and has been changed to the conventional expression of “Pump fluence ($\mu\text{J}/\text{cm}^2$)” in Fig. 1, Fig. 3 and the supplementary materials.

Comment-2: Fig.3b: the authors show spectra with single-mode lasing below a fluence of $40 \mu\text{J}/\text{cm}^2$. However, it would be useful to show the spectra at fluences higher than $40 \mu\text{J}/\text{cm}^2$. Is there any dramatic change to the multimode regime of lasing?

Our response: We appreciate the valuable comment of the reviewer. We agree with the reviewer that lots of the single mode microlasers change to multimode laser at higher pumping fluence in previous experiments. We have carefully checked our samples. We find that the shoulder at high pump fluence comes from the symmetry defect of the sample structure. Based on the reviewer's comment, we have improved our nanofabrication process and solved this problem. With these new samples, the single mode laser operation is well maintained from the laser threshold to gain saturation (see **Fig. R17** below).

Figure R17 (Figure 3 (b) in the main text): The spectra of perovskite VCSELs with $q = 1$ at different pump fluence.

In the revised manuscript, we have replaced the experimental results of perovskite VCSELs and the corresponding discussion in Para-1, Page-5 and Para-1, Page-7 in the main text.

“The single mode operation is well preserved from the laser threshold to the gain saturation.”

“Note that the Archimedean spiral grating also supports the single mode operation. The ration between the laser peak and the photoluminescence is more than 10 dB at the largest pump fluence (see Fig. S21 in the supplementary information).”

Comment-3: It is important to give the fluence values for Figs. 3d, 4, and 5 as well as give corresponding PL spectra. Indeed, there should be information on the ratio between spontaneous PL background and intensity of lasing mode.

Our response: We thank the reviewer for this very careful review and valuable suggestions. Following the reviewer’s suggestion, we have added the pump fluence in Figs. 3d, 4, and 5. We have also added the ratio between PL and intensity of lasing mode in the main text. As shown in **Fig. R18**, the intensity difference is more than 10 dB.

Figure R18 (Figure S21 in the Supplementary Information): The intensity difference between the laser peak and the spontaneous emission background in the old data and the new data, respectively.

In the revised manuscript, we have also added the ratio between the lasing mode and photoluminescence in Para-1, Page-7 of the main text. “Note that the Archimedean spiral grating also supports the single mode operation. The ration between the laser peak and the photoluminescence is more than 10 dB at the largest pump fluence (see Fig. S21 in the supplementary information).” The corresponding experimental results have been added in **Section-6** of the supplementary information.

Comment-4: The etching procedure of the perovskites might affect the PL quantum yield. Thus, it is important to show any comparison of PL properties before and after the etching.

Our response: We thank the reviewer for this valuable suggestion. Since the etching process plays an essential role in the manuscript, the comparison of photoluminescence before and after the etching is very important and necessary. Following the reviewer's suggestion, we have compared the photoluminescence spectra from a grating and a film with the same pump fluence. The results are summarized in **Fig. R19**. It is easy to see that the central peak positions of two spectra are very close. The intensity of photoluminescence from the grating is about 1477. Considering the air region in the grating (~ 0.225), the scaled intensity should be around $1477/0.775=1905$, which is close to the measured value from the film (1935). The slight difference should be induced by the intensity variation of the pump laser. Therefore, we can confirm that the etching process won't degrade the photoluminescence of lead halide perovskites. This is an important basis for this research.

Figure R19: The emission spectra from a grating (blue line) and a film (black line) under the same pump fluence.

In the revised manuscript, we have added the above information in Para-2, Page-3 in the main text. “This change brings two improvements. On one hand, the ion exchange process between Cl⁻ and Br⁻ has been eliminated. The corresponding photoluminescence measurement shows that the emission characteristics have been perfectly preserved during the etching process (see supplementary information Fig. S5). On the other hand, the etching rate can be well controlled to \$\sim 21\$ nm/s with a depth control accuracy of 20 nm (see supplementary information Fig. S6), which enables the shallow etching and will be essential for on-chip applications.”

The corresponding experimental results are summarized in Section 2 of Supplementary Information.

Comment-5: I encourage to explicitly show the image with pumping beam distribution on the gratings to understand how many periods are involved in the process.

Our response: We thank the reviewer for this valuable suggestion. Following the reviewer's suggestion, we have studied the pump profile on the grating. As depicted in **Fig. R20**, the

beam size is about $10\ \mu\text{m}$, which is smaller than the grating size and locates at the center of the grating. Considering the lattice size of the grating, the pump beam roughly covers 36 periods of the grating.

Figure R20: The pump profile on the grating. The beam size is about $10\ \mu\text{m}$, which is smaller than the grating size.

In the revised manuscript, the above information has added the pump beam distribution in the Methods and **Section-6** of the **supplementary information**. “The pump light was focused to a 20 micron in diameter spot on the top surface of the sample through a 40x objective lens. The pump laser beam is roughly half of the perovskite grating (see Fig. S15 in the supplementary information).”

Comment-6: The authors wrote “the material refractive indices are measured by ellipsometer”. However, I did not find the corresponding plot with these values.

Our Response: We thank the reviewer for this very careful review. Following the reviewer’s suggestion, we have added the refractive index of the 300 nm perovskite film in Section 1 of the Supplementary Information. The experimental results are shown in **Fig. R21** below. The corresponding description has been added in Methods of the main text. “The material refractive indices are measured by ellipsometer and shown in Fig. S4 in the supplementary information.”

Figure R21: The reflective index of the 300nm perovskite film as a function of wavelength.

Comment-7: The text quality should be improved. I found a number of typos like “to each other. and two emission”.

Our response: We thank the reviewer for the very careful review. Following the reviewer’s suggestion, we have carefully polished the manuscript. The typos and the grammatical errors have been removed.

Comment-8: In Ref. [4] the authors also observed vortex beam generation from nanopatterned perovskite. Can they compare the current results with their previous work to show the advantages and differences more clearly?

Our response: We thank the reviewer for this valuable comment. The reviewer is correct that Ref. 4 has also reported the vortex laser emission. However, the vortex emission in Ref. [4] is caused by the chiral structure in the momentum space at the bounded states in the continuum (BICs) and the transverse spin angular momentum. This method is strongly dependent on the topological charge (q) at the BICs. The large topological charge of the BICs requires very complicated nanostructures that are quite difficult to be realized in perovskites. In current research, the Archimedean spirals are much easier to be controlled. By tailoring the arm numbers, we have simply changed the topological charge of vortex microlasers from 1 to 4. By further increasing the arm number, the topological charge can even be increased to 32.

In the revised manuscript, we have added the corresponding discussion in Para-2, Page-10 of the main text. “It is important to note that the difference between Archimedean spiral and the bounded states in the continuum (BICs). While both of them can produce vortex laser emission. The topological charge is limited to small value due to the complicated nanostructures. In contrast, the topological charge of Archimedean spiral is determined by the arm number. With the increase of arm number, perovskite VCSELs with topological charge \$q = 8, 16, 32\$ have also been experimentally demonstrated (see Fig. S22 to Fig. S24 in the supplementary information). Such a large range of topological charge makes perovskite

VCSELs essential for high density mode division multiplexing.”

REVIEWERS' COMMENTS:

Reviewer #2 (Remarks to the Author):

In the revised manuscript, the authors present new data and have fabricated new samples to support their claims. I believe that the overall quality of the experimental results is significantly improved. Furthermore, the authors have characterized the robustness and stability of the device along with the emission polarization properties. Although it is still true that a similar scheme was implemented in ref. 10 using organic materials, in the revised manuscript the authors explicitly demonstrate the scalability of their approach by fabricating three additional devices presenting a single-mode emission carrying an $|OAM| = (8, 16, 32)$. This paves the way to high density mode division multiplexing with perovskite-based microlasers. Given the overall quality of the experimental results in the revised manuscript, the detailed analysis of the material and microlaser properties and the experimental demonstration of the scalability of the platform, I believe the manuscript has become rich enough to interest the broad readership of Nature Communications. I support its publication.

Below I append some questions and minor remarks for the authors:

1) Caption of Figure 4 (R2): the authors write "[...](d) The numerically calculated phase profile of emission from the Archimedean spiral Grating [...]". Similar statements can be found in lines 184-185, In the caption of Fig. 5-(d) and in lines 217-218 of the revised manuscript. Could the authors clarify how do they extract these phase maps? In the case the phase maps were directly extracted from the interferograms via a Fourier filtering technique, i.e. they are just a refinement of the experimental data, I would suggest to avoid using "numerically calculated" as it might be misleading. In any case, a sentence explaining how the phase maps have been obtained is necessary.

2) Both In the response to reviewer 1 and 2 (comment 6) the authors write: "The topological charge of 32 is the current record value for on-chip integrated vortex microlasers". Although I agree that the present demonstration of lasing with $|OAM| = 1, \dots, 4, 8, 16, 32$ clearly indicates the possibility to implement high density mode division multiplexing, the authors should be more careful when claiming that they have obtained a record value. See for instance B. Bahari et al., "Topological lasers generating and multiplexing topological light" arXiv:1904.11873 (2019), presenting on-chip microlasers with an emission carrying OAMs of 100, 156 and 276.

3) Caption of Figure 4 (R2): the authors write "[...] (b) and (c) are the simulated self-interference patterns of the laser beams [...]". I guess this is a typo and the authors intend "measured".

4) Line 179: "Different from the Bullseye VCSELs, there are a pair of inverted forks marked by the arrows [...]". However, in Fig. 4-(b,c) no arrow indicating the pitchfork bifurcations is visible, please add them.

5) Lines 130-132: "These unique characteristics make the bullseye VCSELs essential for some particular applications such as biological imaging and optical manipulation". The authors should either be more specific or provide a reference supporting their statement.

6) Line 135: The subject of the sentence is missing. "[...] No obvious reduction (of the laser emission intensity ?) has been observed after 3.6×10^6 continuous pump with a fluence of $37 \mu\text{J}/\text{cm}^2$ (see Fig. S20 in the supplementary information)."

7) There are still several typos in the manuscript. Please double check the revised version.

Sincerely,
Nicola Carlon Zambon

Reviewer #3 (Remarks to the Author):

The authors have addressed all my comments and substantially improved the manuscript quality. Thus, I can recommend it for publication in the current form.

Prof. Sergey Makarov

Reply to Reviewer #2

We thank the reviewer for the very careful review and valuable suggestions. Based on the reviewer's suggestions, we have made the following modification.

Comment-1: Caption of Figure 4 (R2): the authors write “[...](d) The numerically calculated phase profile of emission from the Archimedean spiral Grating [...]”. Similar statements can be found in lines 184-185, In the caption of Fig. 5-(d) and in lines 217-218 of the revised manuscript. Could the authors clarify how do they extract these phase maps? In the case the phase maps were directly extracted from the interferograms via a Fourier filtering technique, i.e. they are just a refinement of the experimental data, I would suggest to avoid using “numerically calculated” as it might be misleading. In any case, a sentence explaining how the phase maps have been obtained is necessary.

Our response: We thank the reviewer for the very careful review and valuable suggestion. We agree with the reviewer that a sentence explaining how the phase maps have been obtained is necessary. In order to obtain the phase maps, we used Lumerical FDTD analysis method to numerically calculate the far field pattern of the nanostructure. The geometry of the nanostructure is taken from the SEM image and electric dipole is used as a point source. After matching the simulated far field laser beam profiles from the perovskite spiral gratings to the experimental results, the phase along the propagating direction is retrieved from simulation. We added a new description about the phase profile “The retrieved phase from the corresponding numerical calculation using Lumerical FDTD analysis shows a phase singularity at the center of the beam (see Fig. 4(d)).” & “These results are retrieved through Lumerical FDTD analysis after matching the simulated result to experiment.” in Para-1, Page-8 and Para-1, Page-10, respectively.

Comment-2: Both In the response to reviewer 1 and 2 (comment 6) the authors write: “The topological charge of 32 is the current record value for on-chip integrated vortex microlasers”. Although I agree that the present demonstration of lasing with $|OAM|=1, \dots, 4, 8, 16, 32$ clearly indicates the possibility to implement high density mode division multiplexing, the authors should be more careful when claiming that they have obtained a record value. See for instance B. Bahari et al., "Topological lasers generating and multiplexing topological light" arXiv:1904.11873 (2019), presenting on-chip microlasers with an emission carrying OAMs of 100,156 and 276.

Our response: We thank the reviewer for the very valuable question. We agree with the reviewer that our demonstration of lasing with $|OAM|=1, \dots, 4, 8, 16, 32$ has not obtained a record value. In the manuscript, we removed the word of record and used “On-chip integrated vortex microlasers with \$q = 0, 1, 2\$, and up to 32 have been experimentally demonstrated.” to describe the result in Para-2, Page-2.

Comment-3: Caption of Figure 4 (R2): the authors write “[...] (b) and (c) are the simulated self-interference patterns of the laser beams [...]”. I guess this is a typo and the authors intend “measured”.

Our response: We thanks the reviewer for the very careful review. We have corrected the typo error to “(b) and (c) are the measured self-interference patterns of the laser beams in the

forward direction and in the backward direction, respectively.” in the caption of Figure 4.

Comment-4: Line 179: "Different from the Bullseye VCSELs, there are a pair of inverted forks marked by the arrows [...]". However, in Fig. 4-(b,c) no arrow indicating the pitchfork bifurcations is visible, please add them.

Our response: We thanks the reviewer for the very careful review and valuable comment. The pitchfork in the experiment in the revised manuscript is very obvious compared to the previous one. In order to ensure the clarity of the picture, we removed the arrow mark. The description in the text has been modified to “Different from the Bullseye VCSELs, there are a pair of inverted forks, which is a direct confirmation of the phase singularity within the Archimedean VCSELs.” in Para-1, Page-8.

Comment-5: Lines 130-132: “These unique characteristics make the bullseye VCSELs essential for some particular applications such as biological imaging and optical manipulation”. The authors should either be more specific or provide a reference supporting their statement.

Our response: We thanks the reviewer for the very careful review and valuable suggestion. Following the reviewer’s suggestion, we have added “Elizabeth A. Munro, Hart Levy, Dene Ringuette, Thomas D. O’Sullivan, and Ofer Levi, Optics Express, 19, 10747 (2011)” as Ref. 46. In this reference, VCSELs have been used as illumination source for simultaneous imaging of blood flow and tissue oxygenation dynamics ex vivo and in vivo and optical imaging of blood flow changes and oxygenation changes in response to induced ischemia has been demonstrated.

Comment-6: Line 135: The subject of the sentence is missing. "[...] No obvious reduction (of the laser emission intensity ?) has been observed after 3.6×10^6 continuous pump with a fluence of $37 \mu\text{J}/\text{cm}^2$ (see Fig. S20 in the supplementary information)."

Our response: We thanks the reviewer for the very careful review. We have changed the sentence to “The perovskite VCSELs also have very nice stability. No obvious reduction of the laser emission intensity has been observed after \$3.6 \times 10^6\$ continuous pump with a fluence of \$37 \mu\text{J}/\text{cm}^2\$ (see Supplementary Figure 20 in the supplementary information).” in Para 1, Page 6.

Comment-7: There are still several typos in the manuscript. Please double check the revised version.

Our response: We thank the reviewer for the very careful review. Following the reviewer’s suggestion, we have carefully polished the manuscript. The typos and the grammatical errors have been removed.